# (-)-Fenchone Ameliorates TNBS-Induced Colitis in Rats via Antioxidant, Immunomodulatory, and Cytoprotective Mechanisms

**DOI:** 10.3390/ph18010018

**Published:** 2024-12-26

**Authors:** Maria Elaine Cristina Araruna, Edvaldo Balbino Alves Júnior, Catarina Alves de Lima Serafim, Matheus Marley Bezerra Pessoa, Michelle Liz de Souza Pessôa, Vitória Pereira Alves, Marianna Vieira Sobral, Marcelo Sobral da Silva, Adriano Francisco Alves, Maria Carolina de Paiva Sousa, Aurigena Antunes Araújo, Leônia Maria Batista

**Affiliations:** 1Postgraduate Program in Natural and Synthetic Bioactive Products, Health Sciences Center, Federal University of Paraíba (UFPB), João Pessoa CEP 58051-970, PB, Brazil; elaine.araruna@gmail.com (M.E.C.A.); edvaldojunioralves@gmail.com (E.B.A.J.); catarinaalvesdelima@gmail.com (C.A.d.L.S.); mmarleybp@gmail.com (M.M.B.P.); michelleliz2008@hotmail.com (M.L.d.S.P.); pereiravitoria689@gmail.com (V.P.A.); mariannavbs@gmail.com (M.V.S.); marcelosobral.ufpb@gmail.com (M.S.d.S.); adriano.alves@academico.ufpb.br (A.F.A.); 2Department of Physiology and Pathology, Health Sciences Center, Federal University of Paraíba (UFPB), João Pessoa CEP 58051-970, PB, Brazil; carolpsousa96@hotmail.com; 3Department of Morphology, Histology and Basic Pathology, Biosciences Center, Federal University of Rio Grande do Norte (UFRN), Natal CEP 59078-970, RN, Brazil; auriprinino@gmail.com

**Keywords:** (-)-fenchone, ulcerative colitis, antioxidant, immunomodulatory, cytoprotection

## Abstract

Background: (-)-Fenchone is a bicyclic monoterpene present in the plant species *Foeniculum vulgare* Mill, *Thuja occidentalis* L. (tuja), and *Lavandula stoechas* (lavender). These plants have therapeutic value in the treatment of intestinal disorders. Aim: To evaluate intestinal anti-inflammatory activity in an acute and chronic trinitrobenzene sulphonic acid (TNBS)-induced colitis model in rats. Methods: Intestinal anti-inflammatory effects were assessed using the acute and chronic TNBS-induced colitis model in rats. The mechanisms were evaluated from colonic tissue fragments of the acute and chronic models. Results: Oral administration of the (-)-fenchone (37.5–300 mg/kg) acute phase or (150 mg/kg) (*p* < 0.001) chronic phase reduced the macroscopic lesion score, ulcerative area, intestinal weight/length ratio, and diarrheal index in TNBS-treated animals. At a dose of 150 mg/kg, the acute and chronic phase decreased malondialdehyde (MDA) and myeloperoxidase (MPO) (*p* < 0.001), restored glutathione (GSH) levels and superoxide dismutase (SOD) (*p* < 0.001), decreased immunomarking for factor nuclear kappa B (NF-κB) and levels of interleukin (IL)-1 and tumor necrosis factor α (TNF-α), and maintained IL-10 and TGF-β basal levels. Furthermore, increased immunostaining for zonula occludens 1 (ZO-1) was observed. Conclusions: (-)-fenchone has intestinal anti-inflammatory activity related to cytoprotection of the intestinal barrier, as well as antioxidant and immunomodulatory effects.

## 1. Introduction

Inflammatory bowel disease (IBD) is characterized by chronic and recurrent intestinal inflammation and tissue remodeling, which primarily affect the gastrointestinal tract [1,2,3], including two main forms called ulcerative colitis (UC) and Crohn’s disease (CD) [2,3].

IBDs are chronic diseases that affect the quality of life of individuals. More than two million Americans are estimated to be living with IBDs, and this population is expected to approach four million by 2030 [4,5]. Although the incidence is stabilizing in many Western countries, the number of cases remains high, as the estimated prevalence exceeds 0.3% of the population in North America and 0.7% in Europe [4,6].

CD is characterized by irregular transmural inflammation that may involve the entire gastrointestinal tract. It is usually associated with complications such as strictures, abscesses, and fistulas [7]. UC, on the other hand, is a non-specific chronic inflammatory disease that affects the innermost lining of the large intestine colon and rectum (submucosal, mucosa) and is characterized by alternating active and remission periods [8]. The pathogenesis of UC is multifactorial, involving genetic abnormalities, altered dietary patterns, intestinal barrier dysfunction, microbiota dysbiosis, and abnormal host immune reactions [9].

The current pharmacotherapy includes the use of corticosteroids, immunosuppressants, 5-aminosalicylates, and biological therapies to reduce the inflammatory process through the immune system and sometimes surgical interventions (corticosteroids, mesalamine compounds, azathioprine, methotrexate, TNF-α, and IL-17A inhibition) [10,11].

The main goal of UC treatment involves inducing and maintaining a long and stable remission, decreasing the incidence of relapses, and preventing further development [8]. However, current therapy fails to prevent relapse of the active phase of IBD; in addition, it is associated with several side effects that contribute to non-adherence to treatment [12,13,14].

In the search for new therapeutic alternatives, natural products have emerged; in the last two centuries, the pharmaceutical industry has used naturally occurring chemical compounds as active principles in and of themselves, as well as a basis for the development of new molecular architectures [15,16]. Terpenes are the most abundant group of secondary metabolites obtained from plants and the main constituents of essential oils [17]. Monoterpenes are natural compounds derived from geranyl diphosphate (GPP) and constitute 90% of essential oils [18]. These compounds have several biological activities already evaluated in experimental models, such as anti-inflammatory [19], antibacterial [20], gastroprotective, and gastric healing [21].

(-)-Fenchone is a bicyclic monoterpene (1,3,3-trimethylbicyclo[2.2.1]heptane-2-one) formed by two isoprene units and an organic ketone function. It is present in the plant species *Foeniculum vulgare* Mill, *Thuja occidentalis* L. (tuja), and *Lavandula stoechas* (lavender) [22,23]. These plants have a therapeutic importance in the treatment of intestinal disorders. Pharmacological studies show that (-)-fenchone exhibits antimicrobial [24], anti-inflammatory [22], antioxidant [23], antiulcerogenic [25], antidiarrheal, and antifungal properties [26].

Previous studies have shown that (-)-fenchone has low acute toxicity LD50 Oral in rats (female) 2.000 mg/kg (OECD Test Guideline 423). Seeking to corroborate previous studies and expand toxicological information, Araruna et al. (2024) [25] evaluated the toxicity with the most effective dose of (-)-fenchone (150 mg/kg) in the acetic-acid-induced gastric ulcer rat model and oral treatment for 14 days. No changes in body mass were noted throughout the treatment. The weights of the organs were not detrimentally affected. In addition, evaluation of biochemical parameters indicated no detrimental effect on hepatic function (AST and ALT) or renal function (urea and creatinine). These findings indicate that 14 days of oral treatment with (-)-fenchone (150 mg/kg daily) has no subacute toxicity in rats [25].

Considering the pharmacological potential of (-)-fenchone and the lack of studies evaluating the effects of these compounds isolated in IBD models, we aimed to evaluate their effects in the trinitrobenzene sulfonic acid (TNBS)-induced ulcerative colitis model.

This study is the first to demonstrate the comprehensive anti-inflammatory, antioxidant, and immunomodulatory effects of (-)-fenchone in both acute and chronic phases of TNBS-induced colitis in rats. The findings provide novel insights into the therapeutic potential of (-)-fenchone as a natural treatment for inflammatory bowel disease, paving the way for further pre-clinical and clinical studies.

## 2. Results

### 2.1. Effect of (-)-Fenchone in Intestinal Acute Inflammation in a TNBS-Induced Colitis Model

The parameters: ulcerative lesion area (ULA), lesion score, weight/length ratio, and diarrhea were evaluated 48 h after TNBS administration, as shown in Table 1. TNBS administration to colitic groups led to the development of large lesions and intense signs of inflammation in gross ulcerative areas. These effects were followed by an increased weight/length ratio of the colonic segment and the diarrhea score, which was determined by the evacuation index (EI) when compared to the non-colitic group (*p* < 0.001). The treatments with (-)-fenchone (37.5, 75, 150, or 300 mg/kg) (*p* < 0.05, *p* < 0.01; *p* < 0.001) or prednisolone (*p* < 0.001) reduced those parameters in all evaluated doses, showing evident signs of disease recovery (representative images in Figure 1). The most effective dose of (-)-fenchone (150 mg/kg) was selected for the following experiments.

#### 2.1.1. Histological Analysis

Histological sections of the colon belonging to the non-colitic, prednisolone, and (-)-fenchone (150 mg/kg) groups were stained with hematoxylin and eosin fenchone (Figure 2A,C,D); the integrity of the mucosa (M) and goblet cells is observed. In the colitic group (5% tween 80) (Figure 2B), intense mucosal destruction is observed, with ulcer formation, which is characterized by organ surface necrosis and intense acute inflammatory reaction (*) and increased leukocyte infiltrate. In sections stained with Masson’s trichrome, it was found that (-)-fenchone (150 mg/kg) (Figure 2H) reduced extracellular matrix deposition (*p* < 0.001) 9.00 (5–19) compared to the colitic group (5% tween 80) who presented scar tissue (∞) in the area of the mucosal ulcer (*p* < 0.001) 30.50 (20–35) (Figure 2F) (Figure 1).

#### 2.1.2. Effect of (-)-Fenchone in the Modulation of Antioxidant and Anti-Inflammatory Properties in Acute Colitis

##### GSH

In determining the antioxidant activity of (-)-fenchone, it was possible to verify that the colitic group (5% tween 80) showed a reduction in GSH levels to 36.7 ± 5.9 nmol of GSH/mg protein compared to the normal group (68.5 ± 4.6 nmol GSH/mg protein). However, administration of (-)-fenchone (150 mg/kg) to 55.9 ± 6.5 nmol GSH/mg protein) restored baseline GSH levels (Figure 3A).

##### SOD

Animals in the colitic group (5% tween 80) demonstrated a significant reduction in the activity of the SOD enzyme to 0.24 ± 0.09 U of SOD/mg of the protein compared to the normal group (2.32 ± 0.37 U SOD/mg of protein). However, SOD activity was significantly increased (*p* < 0.001) with the administration of (-)-fenchone (150 mg/kg) to 2.28 ± 0.29 of SOD/mg of the protein, respectively, when compared to the colitic group (5% tween 80) (Figure 3B).

##### MDA

The colitic group (5% tween 80) showed an increase in MDA levels to 173.2 ± 26.8 nmol MDA/g tissue compared to the normal group (109.5 ± 12.07 nmol MDA/g tissue). However, the administration of prednisolone (2 mg/kg) (109.5 ± 12.43 nmol of MDA/g tissue) or (-)-fenchone (150 mg/kg) (115.2 ± 15.84 nmol MDA/g of tissue) respectively, when compared to the colitic group (Figure 3C).

##### MPO

The results showed that in the colitic group (5% tween 80), there was an increase in MPO levels to 62.06 ± 8.41 units of MPO/g of tissue when compared to the normal group (14.39 ± 3.53 units MPO/g tissue). However, the groups treated with prednisolone (2 mg/kg) or (-)-fenchone (150 mg/kg) significantly reduced MPO levels to 35.28 ± 5.54 and 30.44 ± 6.90 units of MPO/g of tissue, respectively, in relation to the colitic group (Figure 3D).

##### Levels of IL-1β, TNF-α and IL-10

**Next, we evaluated the immunomodulatory effects by measuring the levels of cytokines.** (-)-Fenchone significantly decreased (*p* < 0.001) the levels of IL-1β (pg/mL) to 494.3 ± 31.19 compared to the colitic group (5% tween 80) (1145.0 ± 59.81) (Figure 4A). The TNF-α levels (pg/mL) were also decreased after treatment with (-)-fenchone (215.0 ± 23.58) when compared to the colitic group (5% tween 80) (493.5 ± 47.77) (Figure 4B). Furthermore, (-)-fenchone prevented (*p* < 0.01) the reduction in IL-10 (pg/mL) (251.3 ± 25.88) in comparison to the control group (192.0 ± 19.49) (Figure 4C)

#### 2.1.3. Immunohistochemical Analysis for NF-κB, TGF-β, and ZO-1 in Colon Samples Submitted to Acute Inflammation in a TNBS-Induced Colitis Model

The colons submitted to TNBS and treated with (-)-fenchone showed a focal and concentrated presence of NF-κB (brown color positivity) in the distal enterocytes close to the lumen of the organ, next to the goblet cells, which significantly reduced immunostaining to 18.0 (10.0–25.0) μm^2^ when compared to the colitic group (24.5 (20.0–29.0) μm^2^), in which the presence of NF-κB is irregular and multifocally marked, especially in the ulcer area (Figure 5). In the evaluation of immunostaining for TGF-β, the results showed that treatment with (-)-fenchone significantly increased positive cells marked for TGF-β to 100.5 (201.0–221.0) μm^2^ when compared to the control group (104.0 (85.0–120.0) μm^2^) (Figure 6). In the evaluation of ZO-1, the treatment with fenchone increased the marking to 107.5 (102.0–112.0) μm^2^ (brown color positivity), which was distributed in a regular and multifocal way, similar to the normal group (200.0 (150.0–222.0) μm^2^). In the colitic group, partial and irregular marking is observed in the colonic mucosa (41.5 (25.0–58.0) μm^2^) (Figure 7).

### 2.2. Effect of (-)-Fenchone in the Chronic Phase with Recurrence of the Intestinal Inflammatory Process in a TNBS-Induced Colitis Model

The following parameters: ulcerative lesion area (ULA), lesion score, and weight/length ratio were evaluated 21 days after TNBS administration, as shown in Table 2 (Figure 8). TNBS administration to the colitic groups led to the development of large lesions and intense signs of inflammation with gross ulcerative areas. These effects were followed by an increased weight/length ratio of the colonic segment when compared to the non-colitic group (*p* < 0.001). The treatments with (-)-fenchone (150 mg/kg) (*p* < 0.001) or prednisolone (*p* < 0.001) reduced those parameters.

#### 2.2.1. Histological Analysis

Histological sections of the colons belonging to the non-colitic, prednisolone, and (-)-fenchone groups stained with hematoxylin and eosin are shown in Figure 9A,C,D; the integrity of the mucosa (M) and goblet cells is observed. In the colitic group (5% tween 80) (9B), intense mucosal destruction is observed, with ulcer formation, which is characterized by organ surface necrosis, intense acute inflammatory reaction (*), and increased leukocyte infiltration. In sections stained with Masson’s trichrome, it was found that (-)-fenchone (Figure 9H) reduced extracellular matrix deposition (*p* < 0.001) 8.00 (5–14) compared to the colitic group (5% tween 80) who presented scar tissue (∞) in the area of the mucosal ulcer 26.50 (20–30) (Figure 9F) (Figure 9).

#### 2.2.2. Effect of (-)-Fenchone in the Modulation of Antioxidant and Anti-Inflammatory Properties in Chronic Phase Colitis with Recurrence of the Intestinal Inflammatory Process

##### GSH

In determining the GSH, it was possible to verify that the colitic group (5% tween 80) showed a reduction in GSH levels to 39.8 ± 4.3 nmol of GSH/mg protein compared to the normal group (59.6 ± 5.7 nmol GSH/mg protein). However, administration of (-)-fenchone (150 mg/kg) to 53.5 ± 6.3 nmol GSH/mg protein nearly restored baseline GSH levels (Figure 10A).

##### SOD

Animals in the colitic group (5% tween 80) demonstrated a significant reduction in the activity of the SOD enzyme to 1.39 ± 0.30 U of SOD/mg of protein compared to the normal group (4.88 ± 0.72 U SOD/mg protein). However, SOD activity was significantly increased (*p* < 0.001) with the administration of (-)-fenchone (150 mg/kg) to 5.34 ± 0.68 of SOD/mg of protein, respectively, when compared to the colitic group (5% tween 80) (Figure 10B).

##### MDA

The colitic group (5% tween 80) showed an increase in MDA levels to 175.1 ± 18.1 nmol MDA/g tissue compared to the normal group (83.4 ± 8.21 nmol MDA/g tissue). However, the administration of (-)-fenchone (150 mg/kg) significantly reduced (*p* < 0.001) MDA levels (86.25 ± 12.6 nmol MDA/g of tissue) when compared to the colitic group (Figure 10C).

##### MPO

The results showed that in the colitic group (5% tween 80), there was an increase in MPO levels to 55.41 ± 7.03 units of MPO/g of tissue when compared to the normal group (4.95 ± 0.82 units MPO/g tissue). However, the groups treated with (-)-fenchone (150 mg/kg) significantly reduced (*p* < 0.001) MPO levels to 23.55 ± 2.55 unit of MPO/g of tissue in relation to the colitic group (Figure 10D).

##### Levels of IL-1β, TNF-α, and IL-10

Next, we evaluated the immunomodulatory effects by measuring the levels of cytokines. (-)-Fenchone significantly decreased (*p* < 0.001) the levels of IL-1β (pg/mL) to 896.2 ± 138.3 compared to the colitic group (5% tween 80) (3126.0 ± 317.1) (Figure 11A). TNF-α levels (pg/mL) were also decreased after treatment with (-)-fenchone (*p* < 0.001) (556.3 ± 36.64) when compared to the colitic group (5% tween 80) (790.2 ± 43.60) (Figure 11B). Furthermore, (-)-fenchone prevented (*p* < 0.01) the reduction in IL-10 (pg/mL) (289.3 ± 24.82) in comparison to the control group (213.5 ± 20.83) (Figure 11C).

#### 2.2.3. Immunohistochemical Analysis for NF-κB, TGF-β, and ZO-1 in Colon Samples Submitted to Chronic Phase Colitis with Recurrence of the Intestinal Inflammatory Process

The colons submitted to TNBS and treated with (-)-fenchone showed a focal and concentrated presence of NF-κB in the distal enterocytes close to the lumen of the organ, which significantly reduced immunostaining to 4.00 (2.0–5.0) μm^2^ when compared to the colitic group (11.00 (9.0–14.0) μm^2^) (Figure 12). In the evaluation of immunostaining for TGF-β, the results showed that treatment with (-)-fenchone significantly increased positive cells marked for TGF-β to 12.00 (11.0–15.0) μm^2^ when compared to the control group (2.50 (1.0–5.0) μm^2^) (Figure 13). In the evaluation for ZO-1, the treatment with fenchone increased the marking to 68.00 (64.0–72.0) μm^2^ with a regular and evenly distributed presence of staining in the mucosal colon; in the colitic group, partial and irregular marking is observed in the colonic mucosa (48.5 (35.0–54.0) μm^2^) (Figure 14).

## 3. Discussion

UC is characterized as a non-transmural inflammation, that is, the inflammatory process normally does not affect the full thickness of the intestinal wall, remaining restricted to the mucosa and submucosa. The inflammatory pattern tends to be symmetrical, continuous, and diffuse [7]. In the absence of timely and effective treatment, progressive and cumulative intestinal injury can lead to complications such as strictures, fistulas, loss of intestinal function, surgical interventions, and cancer [2,27].

The experimental model used was the chemical one, using TNBS, which mimics inflammatory events in humans from the induction of T cell reactivity [28]. TNBS administration induces transmural colitis that is driven by a TH1-mediated immune response, which is characterized by the infiltration of CD4^+^ cells, neutrophils, and macrophages into the lamina propria and the secretion of cytokines [29,30]. The model employs intrarectal administration of TNBS hapten, solubilized in ethanol, which breaks the protective colonic mucus barrier, allowing the hapten to penetrate the lamina propria. TNBS will then haptenize the self or microbial proteins/peptides, making them immunogenic, and triggering both innate and adaptive immune responses in the host [28,29,31]. Additionally, the TNBS-induced model can mimic the acute and chronic stages of IBD [30].

In the acute model, (-)-fenchone, at the doses evaluated (37.5, 75, 150, and 300 mg/kg), significantly reduced (*p* < 0.05; *p* < 0.001) ulcerative lesions when compared to colitic, with a significant reduction in diarrhea and in the severity and extent of the lesion which was reflected in the macroscopic lesion score. Diarrhea is an important parameter for evaluating the intestinal anti-inflammatory effect since its increase indicates a loss of the absorptive capacity of the colon [32]. In addition, reports in the literature demonstrate an association between diarrhea and epithelial barrier dysfunction, which occurs through failures in the integrity of gap junctions [33]. The histomorphological evaluation of tissues from rats subjected to (-)-fenchone treatment showed decreased lesion area and inflammatory infiltrate. Similar effects were previously found for monoterpenes *p*-cymene [34] and geraniol [35].

Given the promising results obtained in the acute model of ulcerative colitis, the best dose (150 mg/kg) was selected to investigate its effect in the 21-day chronic ulcerative colitis model with recurrence on the 14th day and to investigate the antioxidant and immunomodulatory effect of (-)-fenchone. The relapsing ulcerative colitis model is considered to mimic this disease in humans and can be used to evaluate new treatments potentially applicable to IBD [36].

In the chronic model with recurrence of the inflammatory process, (-)-fenchone (150 mg/kg) significantly reduced (*p* < 0.001) ulcerative lesions when compared to the colitic group, with a significant reduction in the severity and extent of the lesion, in the weight/weight ratio, length, and macroscopic score of the lesion, thus evidencing the anti-inflammatory effect of the administration during 21 days of treatment. The weight/length ratio is a parameter for evaluating the inflammatory effect, in which the increase in this ratio in the colonic tissues is associated with the respective increase in edema, one of the first markers of inflammation [37].

The histomorphological evaluation confirms the reduction in the inflammatory infiltrate in the group treated with (-)-fenchone. These results are related to a decrease in the migration of inflammatory cells.

Oxidative stress plays an important role in the pathophysiology of inflammation, as lipid peroxidation can exacerbate free radical chain reactions and activate the release of inflammatory mediators [38,39]. Chronic inflammation causes overactivation of the immune system and triggers high levels of ROS that reduce levels of endogenous antioxidants. This process induces the formation of damage in the mucosal layer and bacterial invasion, which in turn stimulates the immune response and contributes to the progression of IBDs [40]. In both the acute and chronic models, (-)-fenchone (150 mg/kg) increased (*p* < 0.001) the GSH and SOD levels compared to the colitic group, restoring baseline levels. SOD is part of the enzymatic antioxidant system that forms the first line of defense against superoxide anions and hydrogen peroxide [41,42]. GSH participates in the non-enzymatic system that protects the integrity and permeability of the cell membrane, promotes the assembly of mucin oligomers, and helps in the maintenance of immune function and the maintenance of protein structure [43].

In addition, the treatment with (-)-fenchone (150 mg/kg) reduced (*p* < 0.001) the levels of MDA and MPO compared to the colitic group; these enzymes participate in the lipid peroxidation process, activating the epithelium, which favors the infiltration of neutrophils, the synthesis and release of cytokines by macrophages, among other mediators that contribute to oxidative stress [43]. Thus, the anti-inflammatory effect found in our study may be related to the inhibition of toxic lipoperoxides and/or due to the promotion of the synthesis of endogenous antioxidants. The monoterpenes carvacrol [44] and menthol [45] have been shown to reduce MPO-mediated lipoperoxidation and increase antioxidant defenses.

ROS are second messengers for the activation of signal transduction pathways that induce inflammation, such as the signaling pathways of mitogen-activated protein kinases (MAPKs) and NF-κB, a transcription factor that binds to gene promoters, targeting and triggering the transcription of inflammatory cytokines and chemokines [46,47]. Treatment with (-)-fenchone in both the acute and chronic phases significantly reduced (*p* < 0.001) immunostaining for NF-κB, as well as reduced levels of inflammatory cytokines (*p* < 0.001) IL-1β and TNF-α; these cytokines participate in the innate immune response and play a key role in disrupting the state of intestinal homeostasis and are involved in the recruitment of granulocytes and increased TH17 activity [11,48].

The main mechanism for controlling responses in the gut is the activation of regulatory T cells (Treg), which play an important role in the maintenance of gastrointestinal homeostasis and secrete immunoregulatory cytokines such as IL-10 and TGF-β; these cytokines suppress macrophage-mediated inflammatory responses and effector T cells [11,49]. In our study, treatment with (-)-fenchone restored levels (*p* < 0.001) of IL-10 and increased immunostaining (*p* < 0.001) for TGF-β when compared to the colitic group. In IBDs, a reduced expression of TGF-β in the mucosa has been observed, which leads to inadequate conditioning of dendritic cells and the perpetuation of the inflammatory state [11].

In our study, treatment with (-)-fenchone in the acute and chronic phases reduced the levels of inflammatory cytokines (TNF-α and IL-1β) considered responsible for amplifying and maintaining chronic inflammation in IBDs and kept the cytokines IL-10 and TGF-β close to baseline, suggesting an immunomodulatory effect. D’Alessio et al. (2013) [50] studied the monoterpene d-limonene and demonstrated that its intestinal anti-inflammatory effect is related to a decrease in TNF-α concentrations via inhibition of the NF-kB pathway. Studies with geraniol [35] demonstrated inhibition of IL-1β and the NF-κB pathway and corroborated this study.

The intestinal epithelium is arranged in a monolayer of columnar epithelial cells, linked through tight junctions [51], and a protective mucous layer covers the outer surface of the epithelium. Defects in the intestinal barrier in individuals with IBD result in an increase in the uptake of luminal antigens through the intestinal epithelium and represent one of the main pathways in the pathogenesis of IBD [52,53]. An increase in immunostaining for ZO-1 was observed for the groups treated with (-)-fenchone, these findings may be related to the increased expression of Mucin type 2 (MUC-2), which can lead to pre-epithelial protection and hinder the development of inflammation. In a study with the *p*-cymene monoterpene [34], a similar result was shown.

## 4. Materials and Methods

### 4.1. Animals

Male Wistar rats (*Rattus norvegicus*) (180−250 g) were provided by the Animal Production Unit (APU) of the Institute for Research on Drugs and Medicines of the Federal University of Paraiba (IPeFarM/UFPB). The animals were acclimated to the conditions of the local bioterium in a temperature of 23 ± 2 °C and a light–dark cycle of 12 h, fed with industrial pellet food and water ad libium. The animals were randomly distributed among the experimental groups. All experimental protocols followed international principles for the study with laboratory animals [54] and were approved by the Institutional Ethics Commission on Animal Use (CEUA/UFPB) under registration number: 7216040119/19. All efforts were made to reduce the number of animals used, their pain, suffering, and stress. The visual observers who carried out the macroscopic and microscopic analysis were blinded from the identification of the experimental groups.

### 4.2. Reagents

The following substances were used in this study: (-)-fenchone (SIGMA Chemical Co., St. Louis, MO, USA) solubilized in 5% tween 80; sodium acetate (SIGMA Chemical Co., St. Louis, MO, USA); acetonitrile (SIGMA Chemical Co., St. Louis, MO, USA); 5,5′-ditiobis-2-nitrobenzoic acid (DTNB) (SIGMA Chemical Co., St. Louis, MO, USA); eth-ylenediaminetetraacetic acid (EDTA) (SIGMA Chemical Co., St. Louis, MO, USA); bovine serum albumin (BSA) (SIGMA Chemical Co., St. Louis, MO, USA); trichloroacetic acid (SIGMA Chemical Co., St. Louis, MO, USA); trinitrobenzene sulfonic acid (TNBS) (SIGMA Chemical Co., St. Louis, MO, USA); antirat antibodies for IL-1β, TNF-α, and IL-10 (R&D systems); biotinylated sheep polyclonal antibodies (anti-IL-1β, anti-TNF-α, or anti-IL-10) (R&D Systems); immunohistochemical polyclonal rat antibodies (NF-κB or TGF-β) (Cloud-clone Corp); sodium carbonate (MERK, Darmstadt, DE-HE, Germany); potassium chloride (SIGMA Chemical Co., USA); magnesium chloride (SIGMA Chemical Co., St. Louis, MO, USA); sodium chloride PA (QUIMEX-MERCK, Uberaba, MG, Brazil); ethyl ether (MERK, Darmstadt, DE-HE, Germany); monobasic sodium phosphate (SIGMA Chemical Co., St. Louis, MO, USA); bibasic sodium phosphate (SIGMA Chemical Co., St. Louis, MO, USA); prednisolone (SIGMA Chemical Co., St. Louis, MO, USA); Methyl-Phenylindole (SIGMA Chemical Co., St. Louis, MO, USA); ketamine 5% (VETANARCOL); Tris Buffer (Vetec^®^); Trizma Buffer (SIGMA Chemical Co., St. Louis, MO, USA); and xylazine 2% (DORCIPEC).

### 4.3. TNBS-Induced Intestinal Acute Inflammation in Rats

The experimental protocol described by Morris et al. (1989) [55] was conducted with some modifications. After fasting for 24 h, rats were anesthetized with 2% xylazine hydrochloride and 5% ketamine hydrochloride for rectal administration of TNBS (10 mg per animal dissolved in 0.25 mL of 50% ethanol) with the aid of a 2 mm diameter probe, which was inserted around 8 cm inside the rectum. Following the administration of TNBS, animals were kept for 10 min upside down. Each group of animals was pre-treated using an oral gavage with the vehicle 5% tween 80 (colitic group) and 2 mg/kg prednisolone (standard control group). (-)-fenchone (37.5, 75, 150, and 300 mg/kg) was solubilized in tween 5% 80, 48, 24, and 1 h before TNBS administration and 24 h after inflammation induction. Then, after 48 h of TNBS administration, the rats were anesthetized and euthanized. Colonic segments were collected and photographed for ulcerative area (UA) quantification and macroscopic score determination. General parameters such as diarrhea and colon weight/length ratio were also evaluated. An untreated non-colitic group, whose animals were not submitted to inflammation induction, was added to the experiment. Intestinal injury score was assessed according to a scale previously described by Bell et al. (1996) [56]: (1) focal hyperemia, no ulcer; (2) ulceration, no hyperemia/bowel wall thickening; (3) ulceration, inflammation at one site; (4) ulceration, inflammation at 2 or more sites; (5) major injury > 1 cm; 6–10 major damage > 2 cm.

The percentage of injury inhibition (%) was calculated as described:$$\%\;Lesion\;inhibition=\left[\frac{Sample\;UA\times 100}{Colitic\;group\;UL}\right]-100$$

To calculate the weight/length ratio parameter, the following formula was used:$$\frac{Colon\;weight\;(g)}{Colon\;length\;(cm)}$$

Diarrheal status was determined from the evacuation index (EI) for 24 h until euthanasia. Stools were counted and classified as solids, semisolids, and liquids. The EI was calculated as follows:EI = S (no of solid stools × 1) + (no of watery stools × 2) + (no of liquid stools × 3)

### 4.4. Chronic Phase with Recurrence of the Intestinal Inflammatory Process in TNBS-Induced Intestinal Inflammation in Rats

The experimental protocol described by Morris et al. (1989) [55] was conducted with some modifications. After fasting for 24 h, the rats were anesthetized with 2% xylazine hydrochloride and 5% ketamine hydrochloride for rectal administration of TNBS (10 mg per animal dissolved in 0.25 mL of 50% ethanol) with the aid of a 2 mm diameter probe, which was inserted around 8 cm inside the rectum. Following the administration of TNBS, the animals were kept for 10 min upside down. Twenty-four hours after the initial induction, the groups of animals were treated orally with 5% tween 80 (colitic group), 2 mg/kg prednisolone (standard control group), or (-)-fenchone 150 mg/kg. The animals were treated once a day for 21 days. On the 14th day after the first induction, the second administration (recurrence) of TNBS (10 mg/0.25 mL 50% *v*/*v*, rectally) was performed to mimic the recurrent relapses in inflammatory bowel diseases in humans. On the 21st day, the rats were anesthetized and euthanized. Colonic segments were collected and photographed for ulcerative area (UA) quantification and macroscopic score determination. General parameters such as diarrhea and colon weight/length ratio were also evaluated. An untreated non-colitic group, whose animals were not submitted to inflammation induction, was added to the experiment. The intestinal injury score was assessed according to a scale previously described by Bell et al. (1996) [56]: (1) focal hyperemia, no ulcer; (2) ulceration, no hyperemia/bowel wall thickening; (3) ulceration, inflammation at one site; (4) ulceration, inflammation at 2 or more sites; (5) major injury > 1 cm; 6–10 major damage > 2 cm.

After ULA determination, tissue samples of colon injuries were collected, stored, and preserved in a 10%-buffered formaldehyde solution for histological and immunohistochemical analysis. For biochemical assays, tissues were stripped, weighed, and frozen at −80 °C.

#### 4.4.1. Histological Analysis and Morphometric Analysis

Fragments of colon tissue from the TNBS-induced ulcerative colits acute and recurrence-phase protocol from the cysteamine-induced ulcer protocol were preserved in a 10% buffered formaldehyde solution until histological processing. The tissues were embedded in histological paraffin and sectioned with a thickness of 4 µm. From the sectioned tissues, 2 stains were performed: hematoxylin and eosin (HE) and Masson’s trichrome (MT). The histological sections stained by Masson’s trichrome were visualized through the 40x objective of the Olympus microscope (Tokyo, Japan) and 20 random images were digitalized through the same microscope and Q-Color3 microscope, in each image, all pixels with shades of blue (Masson’s trichrome) were selected to create a binary image, digitally processed and calculate the area in μm^2^.

#### 4.4.2. Determination of Reduced Glutathione (GSH), Malondialdehyde (MDA), Myeloperoxidase (MPO), and Superoxide Dismutase (SOD)

##### GSH Determination

The protocol used by Faure and Lafond (1995) [57] was followed to determine the GSH levels. The tissue samples were suspended in 0.02 M 1:10 (*v*/*v*) EDTA. From this homogenate, 400 µL was removed, and 320 µL of distilled water and 80 µL of 50% trichloroacetic acid were added, being centrifuged at 3000 rpm at 4 °C for 15 min. Then, 100 µL of the resulting supernatant was pipetted into a 96-well microplate, and 200 µL of Tris and 25 µL of DTNB were added. This microplate was incubated at room temperature and after 15 min, a reading on a spectrophotometer (Polaris) was performed at a wavelength of 412 nm. The calibration curve was made with reduced L-glutathione. The GSH values of the samples were calculated by interpolating the values with the standard curve and expressed in nmol GSH/mg of protein.

##### MDA Determination

The samples were suspended in Trizma^®^ buffer (Tris HCl) 1:5 (*w*/*v*), homogenized, and centrifuged at 11,000 rpm at 4 °C for 10 min. Soon after, 300 μL of the supernatants was transferred to Eppendorf tubes; 750 μL of the chromogenic compound (10.3 mM of 1-methyl-2-phenylindol) and 225 μL of hydrochloric acid (37%) were added, and incubated at 45 °C in a water bath for 40 min and then were centrifuged again at 11,000 rpm at 4 °C for 5 min. Then, 300 μL of the supernatant was transferred to a 96-well microplate, and absorbance was determined by colorimetry (586 nm), using a plate reader (Polaris’ spectrophotometer). The data were interpolated with the standard curve, and the results were expressed as nmol MDA/g tissue [58].

##### MPO Determination

The tissue sample fragments were homogenized in the hexadecyltrimethylammonium bromide (HTAB) buffer, which has a detergent function, lysing the granules of the neutrophils that contain the myeloperoxidase. After homogenization, the material was subjected to sonication for 5 min. Then, the sample was subjected to a double freezing and thawing process to facilitate the disruption of cellular structures and, consequently, the release of the enzyme. The homogenate was centrifuged at 5,000 rpm at 4 °C for 20 min and concentrated for 24 h. On the following day, 7 μL of the supernatant was collected, to which 200 μL of the reading solution (o-dianisidine hydrochloride, potassium phosphate buffer, and 1% H_2_O_2_) was added. The reading was performed using a spectrophotometer at 450 nm wave, at times 0 and 1 min. The results were expressed as units of myeloperoxidase per gram of tissue [59].

##### SOD Determination

The tissue samples were homogenized in phosphate buffer (0.4 M, pH 7.0) and centrifuged for 15 min at 10,000 rpm at 4 °C. The supernatant was removed and used in the assay. The plates containing the reaction medium (10 mM phosphate buffer), L-methionine (1.79 mg/mL, pH 7.8), riboflavin (0.2 mg/mL, pH 7.8), NBT (1.5 mg/mL, pH 7.8), and 10 μL of the sample supernatant were exposed to a fluorescent lamp (15 W) for 10 min. After this period, the material was taken to the 630 nm spectrophotometer [60].

#### 4.4.3. Immunomodulatory Activity

##### Determination of IL-1, TNF-α, and IL-10 Levels

The levels of pro-inflammatory (IL-1β and TNF-α) and immunoregulatory (IL-10) cytokines were determined using an ELISA immunoenzymatically assay (sandwich type). The capture antibodies for each interleukin were sensitized in a 96-well microplate (flat bottom). After 18 h, the plate was washed with 0.05% tween 20 solution (wash buffer), blocked with a 1% bovine serum albumin solution, and washed with the wash buffer. Soon after, a tissue macerate was prepared in phosphate-buffered saline (PBS) in the proportion of 100 mg of tissue to 600 μL of PBS, homogenized, and centrifuged at 4000 rpm for 10 min at 4 °C. The supernatants (100 μL) were pipetted in a 96-well plate and made the standard curve. The biotinylated secondary antibody (100 μL) was added to each well, followed by incubation for 2 h and 3 washes. The plate was incubated with streptavidin for 20 min, washed 3 times, and the substrate for development was added (DuoSet Kit^©^-R & D Systems Catalog-DY999, Minneapolis, MN, USA) and incubated for 20 min. After this time, the reaction was stopped by adding 50 μL of the stop solution, and the reading was performed on a spectrophotometer at 450 nm. The results were obtained by interpolation with the standard curve and expressed in pg/mL [60].

#### 4.4.4. Immunohistochemical Analysis

Colon sections (5 μm) of 5 animals per group were obtained with a microtome, transferred to silanized slides (Dako, Glostrup Denmark), and subjected to dewaxing and hydration. Then, the slides were subjected to the simple indirect method of blocking with Anti-Peroxidase Peroxidase (APP), where the primary antibody is directed against the antigen (protein) to be detected, and the secondary antibody serves as a bridge for binding to the APP complex. Then, the slides were incubated overnight at 4 °C with the following primary antibodies a: NF-κB, TGF-β, and ZO-1. After being washed with distilled water, the slides were incubated with a secondary for 60 min, and 3,3′-diaminobenzidine (DAB, Biocare Medical, Concord, CA, USA) was used as the chromogen; the specimens were counterstained with hematoxylin. The samples were visualized under an optical microscope (Olympus microscope, Tokyo, Japan) coupled to a camera (Nikon DS-Ri2, Melville, NY, USA).

### 4.5. Statistical Analysis

Data were expressed as mean ± standard deviation (SD) or mean ± standard error (SE) for parametric values or expressed as median (minimum value and maximum value) for non-parametric data. One-way ANOVA (parametric data) or Kruskal–Wallis test (non-parametric data) was performed, followed by Dunnet and Tukey or Dunn post-tests, respectively. The results were considered significant when *p* < 0.05. All data were analyzed using GraphPad^®^ 5.0 software.

## 5. Conclusions

Our data demonstrated that the monoterpene (-)-fenchone has intestinal anti-inflammatory activity related to antioxidant (increase in GSH and SOD with a reduction in MDA and MPO) and immunomodulatory (reduction in NF-κB, TNF-α, and IL-1β and increase in IL-10 and TGF) effects, as well as cytoprotective mechanisms, acting on the pre-epithelial or epithelial barrier. Thus, these results suggest a potential use of this substance for the treatment of intestinal inflammation.

Additional studies should be conducted to further elucidate the molecular mechanisms and signaling pathways by which (-)-fenchone exerts its anti-inflammatory and antioxidant effects.

## Figures and Tables

**Figure 1 pharmaceuticals-18-00018-f001:**
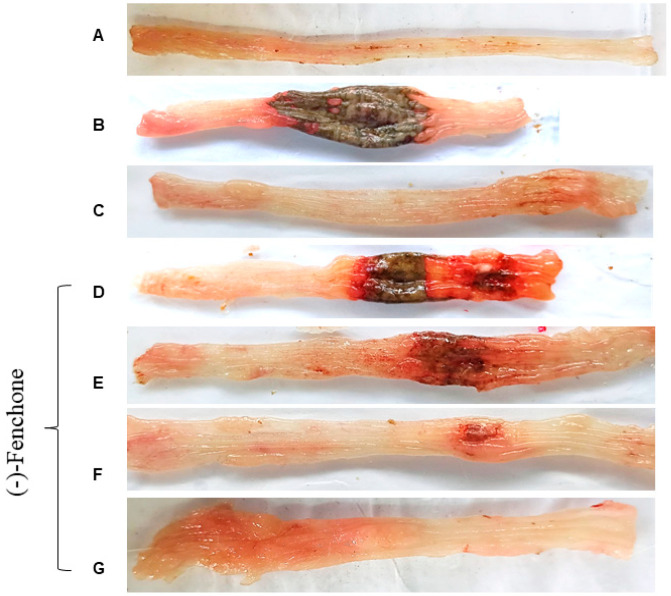
Representative images of rat colons from non-colitic group (**A**); colitic groups (5% tween 80) (**B**); prednisolone, 2 mg/kg (**C**); and (-)-fenchone, 37.5 mg/kg (**D**), 75 mg/kg (**E**), 150 mg/kg (**F**), and 300 mg/kg (**G**) subjected to acute TNBS-induced ulcerative colitis model.

**Figure 2 pharmaceuticals-18-00018-f002:**
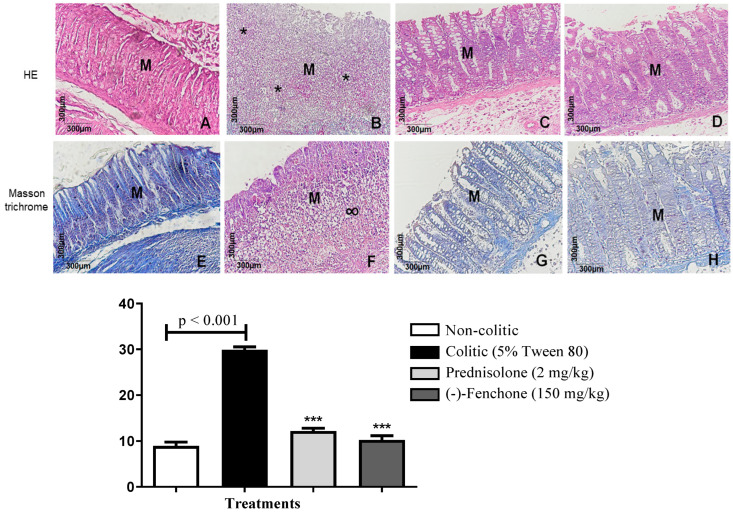
Microscopic effects of the colon of rats submitted to acute TNBS-induced ulcerative colitis model and treated or not with (-)-fenchone. Representative photomicrographs of the animals’ colons from the experimental groups: HE staining of non-colitic (**A**), colitic (5% tween 80) (**B**), prednisolone 2 mg/kg (**C**), and (-)-fenchone 150 mg/kg (**D**). Masson trichrome—non-colitic (**E**), colitic (5% tween 80) (**F**), prednisolone 2 mg/kg (**G**), and (-)-fenchone 150 mg/kg (**H**). Colonic mucosa (M), acute inflammatory reaction (*), scarring tissue (∞). Results are expressed as the median (minimum–maximum) of the parameters analyzed (n = 5, 3 sessions per animal). Kruskal–Wallis test and Dunn’s posterior test were performed using the software Graph Pad Prism 6. *** *p* < 0.001 compared to the non-colitic group (5% tween 80).

**Figure 3 pharmaceuticals-18-00018-f003:**
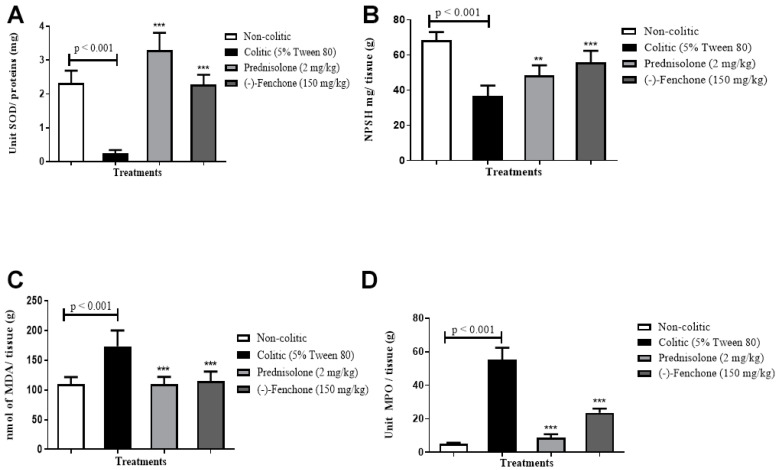
Effect of oral administration of (-)-fenchone and prednisolone in GSH (**A**), SOD (**B**), MDA (**C**), and MPO (**D**) levels from acute TNBS-induced ulcerative colitis model in rats. Results are expressed as mean ± SE of the parameters analyzed (n = 5–8). The one-way ANOVA followed by Dunnett’s and Tukey’s test. ** *p* < 0.01 or *** *p* < 0.001 vs. non-colitic group.

**Figure 4 pharmaceuticals-18-00018-f004:**
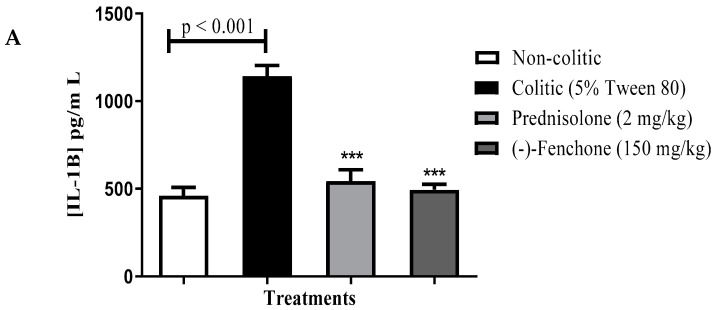
Effect of (-)-fenchone and prednisolone in IL-1β (**A**), TNF-α (**B**), and IL-10 (**C**) levels from acute TNBS-induced ulcerative colitis model in rats. Results are expressed as mean ± SE of the parameters analyzed (n  =  6–8). The analysis of variance of one via (ANOVA) followed by Dunnett’s and Tukey’s test. ** *p* < 0.01 or *** *p* < 0.001 vs. control group.

**Figure 5 pharmaceuticals-18-00018-f005:**
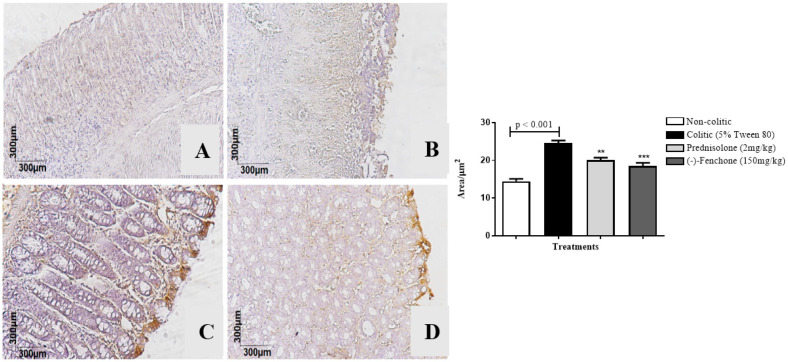
Photomicrographs with immunohistochemical marking for nuclear transcription factor kappa B (NF-κB) in colon samples from rats. Non-colitic (**A**); colitic (5% tween 80) (**B**); prednisolone (2 mg/kg) (**C**); (-)-fenchone (150 mg/kg) (**D**). Results are expressed as the median (minimum–maximum) of the parameters analyzed (n = 5, 3 sessions per animal). Kruskal–Wallis test and Dunn’s posterior test were performed using the software Graph Pad Prism 6. ** *p* < 0.01, *** *p* < 0.001 compared to the colitic group (5% tween 80).

**Figure 6 pharmaceuticals-18-00018-f006:**
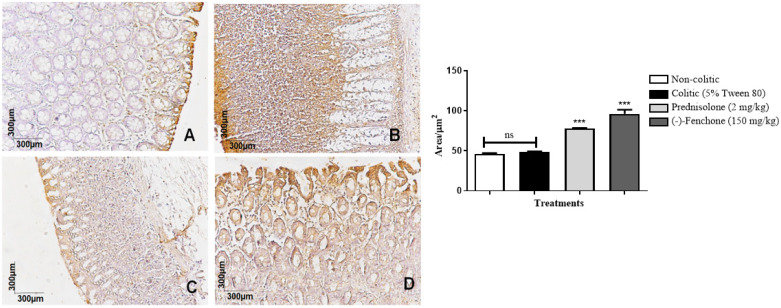
Photomicrographs with immunohistochemical marking for transforming growth factor beta (TGF-β) in colon samples from rats. Non-colitic (**A**); colitic (5% tween 80) (**B**); Prednisolone (2 mg/kg) (**C**); (-)-fenchone (150 mg/kg) (**D**). Results are expressed as the median (minimum–maximum) of the parameters analyzed (n = 5, 3 sessions per animal). Kruskal–Wallis test and Dunn’s posterior test were performed using the software Graph Pad Prism 6. *** *p* < 0.001 compared to the colitic group (5% tween 80).

**Figure 7 pharmaceuticals-18-00018-f007:**
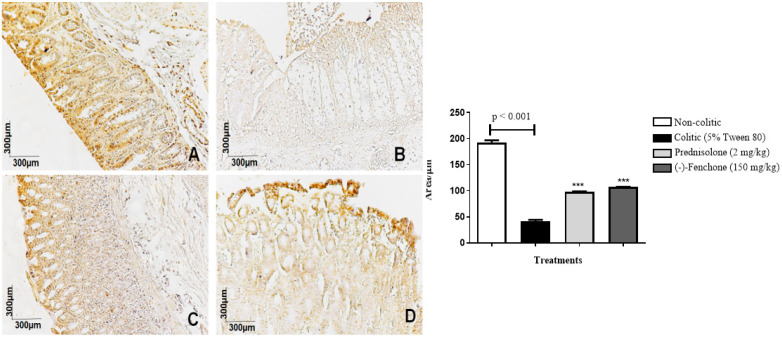
Photomicrographs with immunohistochemical marking for zonula occludens 1 (ZO-1) in colon samples from rats. Non-colitic (**A**); colitic (5% tween 80) (**B**); prednisolone (2 mg/kg) (**C**); (-)-fenchone (150 mg/kg) (**D**). Results are expressed as the median (minimum–maximum) of the parameters analyzed (n = 5, 3 sessions per animal). Kruskal–Wallis test and Dunn’s posterior test were performed using the software Graph Pad Prism 6. *** *p* < 0.001 compared to the colitic group (5% tween 80).

**Figure 8 pharmaceuticals-18-00018-f008:**
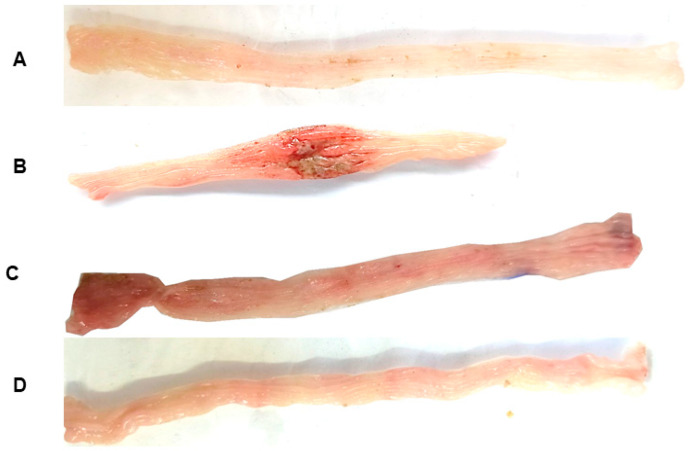
Representative images of rat colons from non-colitic group (**A**); colitic groups (5% tween 80) (**B**); prednisolone 2 mg/kg (**C**); (-)-fenchone 150 mg/kg (**D**) subjected to sub-chronic phase with recurrence of the intestinal inflammatory process in a TNBS-induced colitis model.

**Figure 9 pharmaceuticals-18-00018-f009:**
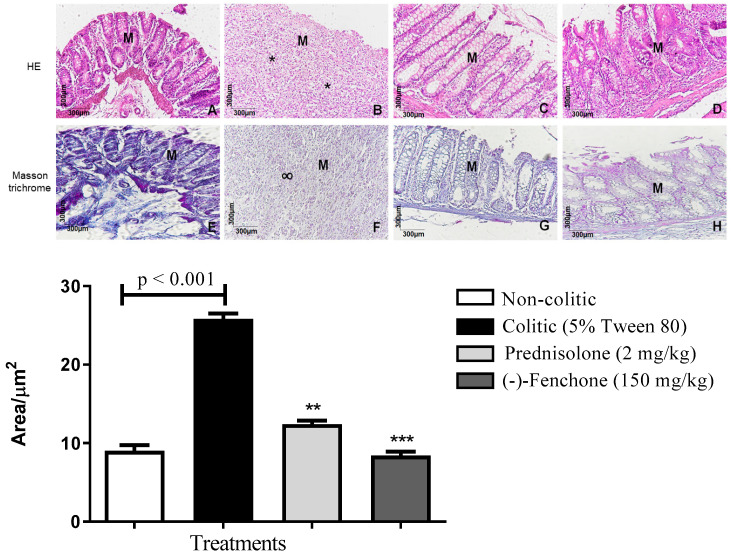
Microscopic effects of the colon of rats submitted to in sub-chronic phase with recurrence of the intestinal inflammatory process in a TNBS-induced colitis model and treated or not with (-)-fenchone. Representative photomicrographs of the animals’ colon from the experimental groups: HE staining: non-colitic (**A**), colitic (5% tween 80) (**B**), prednisolone 2 mg/kg (**C**) and (-)-fenchone 150 mg/kg (**D**). Masson trichrome: non-colitic (**E**), colitic (5% tween 80) (**F**), prednisolone 2 mg/kg (**G**), and (-)-fenchone 150 mg/kg (**H**). Colonic mucosa (M), acute inflammatory reaction (*), scarring tissue (∞). Results are expressed as the median (minimum–maximum) of the parameters analyzed (n = 5, 3 sessions per animal). Kruskal–Wallis test and Dunn’s posterior test were performed using the software Graph Pad Prism 6. ** *p* < 0.01 or *** *p* < 0.001 compared to the non-colitic group (5% tween 80).

**Figure 10 pharmaceuticals-18-00018-f010:**
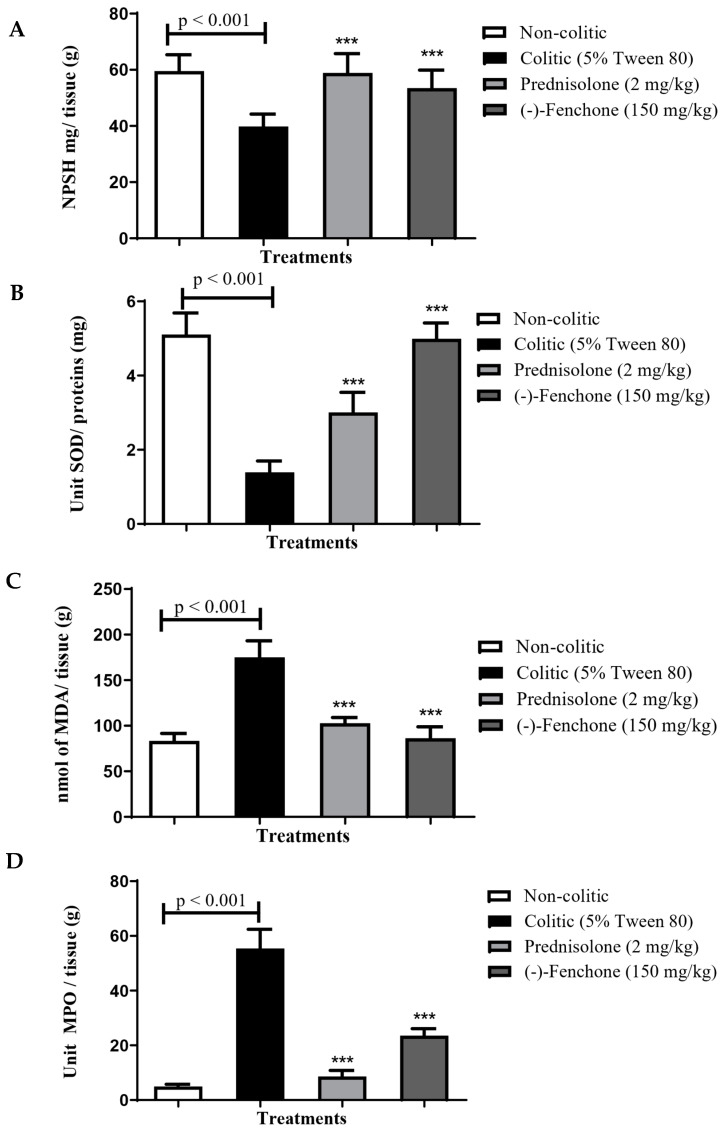
Effect of oral administration of (-)-fenchone and prednisolone in A GSH (**A**), SOD (**B**), MDA (**C**), and MPO (**D**) levels form in sub-chronic phase with recurrence of the intestinal inflammatory process in a TNBS-induced colitis model in rats. Results are expressed as mean ± SE of the parameters analyzed (n = 5–8). The one-way analysis of variance (ANOVA) followed by Dunnett’s and Tukey’s test. *** *p* < 0.001 vs. non-colitic group.

**Figure 11 pharmaceuticals-18-00018-f011:**
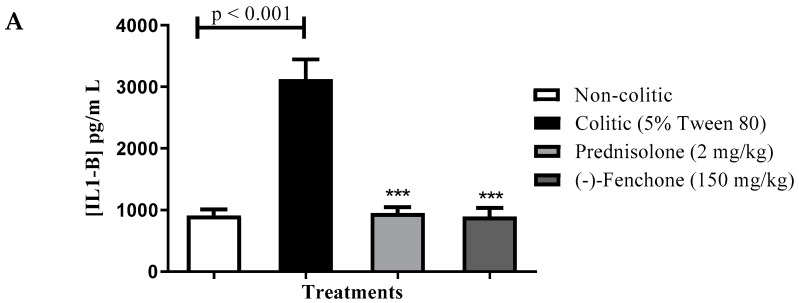
Effect of (-)-fenchone and prednisolone in IL-1β (**A**), TNF-α (**B**), and IL-10 (**C**) levels form in sub-chronic phase with recurrence of the intestinal inflammatory process in a TNBS-induced colitis model in rats. Results are expressed as mean ± SE of the parameters analyzed (n  =  6–8). The one-way analysis of variance (ANOVA) followed by Dunnett’s and Tukey’s test. ** *p* < 0.01 or *** *p* < 0.001 vs. control group.

**Figure 12 pharmaceuticals-18-00018-f012:**
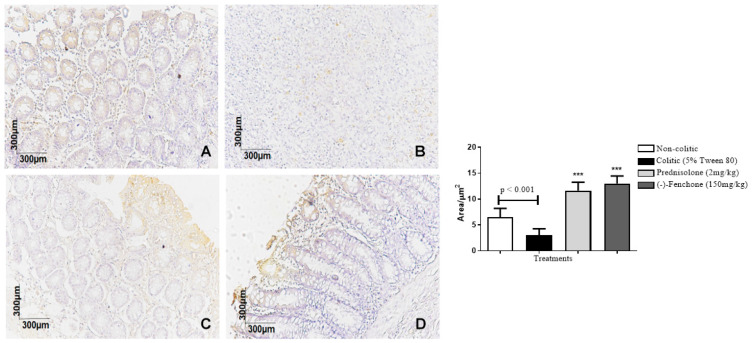
Photomicrographs with immunohistochemical marking for transforming growth factor beta (TGF-β) in colon samples from in sub-chronic phase with recurrence of the intestinal inflammatory process in a TNBS-induced colitis model in rats. Non-colitic (**A**); colitic (5% tween 80) (**B**); prednisolone (2 mg/kg) (**C**); (-)-fenchone (150 mg/kg) (**D**). Results are expressed as the median (minimum–maximum) of the parameters analyzed (n = 5, 3 sessions per animal). Kruskal–Wallis test and Dunn’s posterior test were performed using the software Graph Pad Prism 6. *** *p* < 0.001 compared to the colitic group (5% tween 80).

**Figure 13 pharmaceuticals-18-00018-f013:**
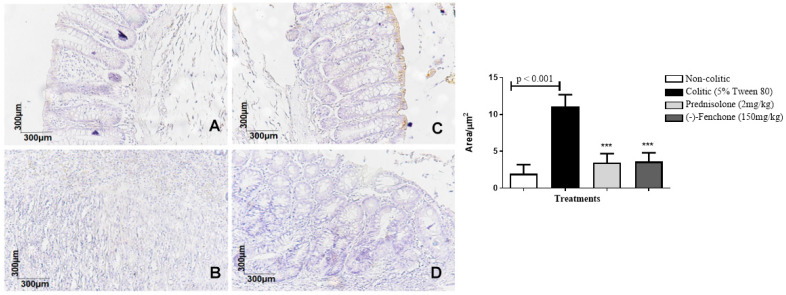
Photomicrographs with immunohistochemical marking for nuclear transcription factor kappa B (NF-κB) in colon samples from in sub-chronic phase with recurrence of the intestinal inflammatory process in a TNBS-induced colitis model in rats. Non-colitic (**A**); colitic (5% tween 80) (**B**); prednisolone (2 mg/kg) (**C**); (-)-fenchone (150 mg/kg) (**D**). Results are expressed as the median (minimum–maximum) of the parameters analyzed (n = 5, 3 sessions per animal). Kruskal–Wallis test and Dunn’s posterior test were performed using the software Graph Pad Prism. *** *p* < 0.001 compared to the colitic group (5% tween 80).

**Figure 14 pharmaceuticals-18-00018-f014:**
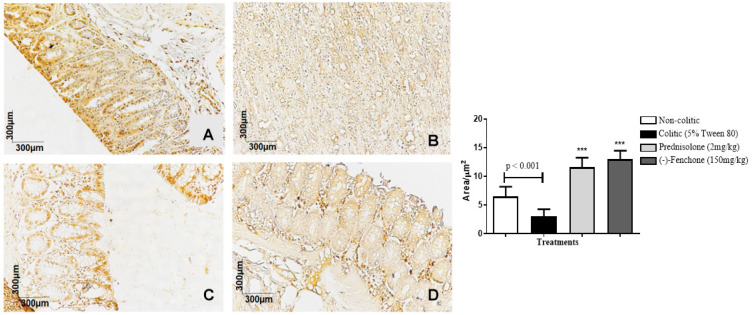
Photomicrographs with immunohistochemical marking for zonula occludens 1 (ZO-1) in the colon samples from the sub-chronic phase with recurrence of the intestinal inflammatory process in the TNBS-induced colitis model in rats. Non-colitic (**A**); colitic (5% tween 80) (**B**); prednisolone (2 mg/kg) (**C**); (-)-fenchone (150 mg/kg) (**D**). Results are expressed as the median (minimum–maximum) of the parameters analyzed (n = 5, 3 sessions per animal). Kruskal–Wallis test and Dunn’s posterior test were performed using the software Graph Pad Prism 6. *** *p* < 0.001 compared to the colitic group (5% tween 80).

**Table 1 pharmaceuticals-18-00018-t001:** Effect of (-)-fenchone in intestinal acute inflammation in a TNBS-induced colitis model.

Group	Dose (mg/kg)	ULA (mm^2^)	% Lesion Inhibition	Lesion Score	Weight/ Length (mg/cm)	Evacuation Index (EI)
Non-colitic	-	ND	ND	ND	74.9 ± 5.9	12.0 ± 3.0
Colitic (5% tween 80)	-	286.6 ± 5.7 ^###^	-	6.0 ± 0.9 ^###^	162.7 ± 22.4 ^###^	20.0 ± 2.4 ^#^
Prednisolone	2	36.7 ± 5.7 ***	87%	3.0 ± 0.5 ***	107.5 ± 15.0 ***	7.0 ± 3.0 ***
(-)-Fenchone	37.5	240.9 ± 24.9 *	16%	4.5 ± 1.2 *	126.7 ± 10.9 *	9.0 ± 1.1 *
	75	181.4 ± 22.7 ***	37%	4.0 ± 1.5 *	118.5 ± 32.9 **	7.0 ± 0.7 ***
	150	72.2 ± 10.0 ***	75%	3.0 ± 0.7 **	113.1± 18.7 **	6.0 ± 1.1 ***
	300	73.8 ± 10.8 ***	74%	3.0 ± 0.8 **	112.8 ± 9.7 **	4.0 ± 1.3 ***

Data are expressed as mean ± SD or median (minimum/maximum values). For parametric data, we used one-way ANOVA followed by Dunnett’s and Tukey’s post-tests. For non-parametric data, we used the Kruskal–Wallis test and Dunn’s test. * *p* < 0.05, ** *p* < 0.01, *** *p* < 0.001 compared to colitic groups; ^#^
*p* < 0.05,^###^
*p* < 0.001 compared to non-colitic group. (*n* = 7–10). ND = not detectable.

**Table 2 pharmaceuticals-18-00018-t002:** Effect of (-)-fenchone in sub-chronic phase with recurrence of the intestinal inflammatory process in a TNBS-induced colitis model.

Groups	ULA (mm^2^)	% Lesion Inhibition	Lesion Score	Weight/Length (mg/cm)
Non-colitic	ND	ND	ND	91.4 ± 3.5
Colitic (5% tween 80)	57.17 ± 7.5 ^###^	0	3.0 ± 0.7 ^###^	148.9 ± 10.5 ^###^
Prednisolone (2 mg/kg)	11.00 ± 2.1 ***	81%	1.0 ± 0.6 ***	127.5 ± 13.7 **
(-)-fenchone (150 mg/kg)	11.50 ±2.9 ***	80%	1.5± 0.7 **	130.6 ± 8.3 **

Data are expressed as mean ± SD or median (minimum/maximum values). For parametric data, we used one-way ANOVA followed by Dunnett’s and Tukey’s post-tests. For non-parametric data, we used the Kruskal–Wallis test and Dunn’s test. ** *p* < 0.01, *** *p* < 0.001 compared to colitic groups; ^###^
*p* < 0.001 compared to non-colitic group. (n = 7–10). ND = not detectable.

## Data Availability

The original contributions presented in this study are included in the article. Further inquiries can be directed to the corresponding author.

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
