# Peer review of "(-)-Fenchone Ameliorates TNBS-Induced Colitis in Rats via Antioxidant, Immunomodulatory, and Cytoprotective Mechanisms"

_pharmaceuticals, 2024, doi:10.3390/ph18010018_

Round 1
Reviewer 1 Report
Comments and Suggestions for Authors
There is no sign of damage on the figures. Added
Also, the resolution of Figures 6-7 should be increased.
I could not notice the novelty of the article. The authors should express it more clearly in the introduction section.
Spelling errors should be reviewed.
There is a bit too much general information in the Discussion. It would be better if there were more comparisons with the literature.
Conclusion is very short, the possible effects that it will present to future studies should be mentioned.
Comments on the Quality of English LanguageGood.
Author Response
- There are no signs of damage to the figures. Added
The damage was evident in Figures 1B, colitic group, and Figures 1D, 1E, and damage was reduced with increased doses of (-)-Fenchone (1F, 1G), according to values ​​in Table 1. As in Figure 8B, the colitic group, as values ​​in Table 2. Figures 2B and 8B show the inflammatory process in the colitic group (marked by *) in the histological images of slides stained with hematoxylin and eosin.
- Furthermore, the resolution of Figures 6-7 should be increased.
The figures have been enlarged.
- I didn't understand how new the article was. The authors should express this more clearly in the introduction section.
Information about the objective and novelty of the article was added in the introduction.
- Spelling errors must be proofread.
Language and spelling errors have been revised.
- There is a little too much general information in the discussion. It would be better if there were more comparisons with literature.
The discussion has been revised with more concise information.
- The conclusion is very short, and it is worth mentioning the possible effects it will present for future studies.
The conclusion has been rewritten with additional information.
Reviewer 2 Report
Comments and Suggestions for Authors
Fenchone ameliorate(s)TNBS-induced colitis in rats via antioxidant, immunomodulatory and cytoprotective mechanisms, examines the ability of a terpene to relieve TNBS damage to rat colon in both an acute setting and chronically. The study looks at a variety of parameters to make a case for the effectiveness of the agent. The use of a steroid as a comparison is very good. In general, the work demonstrates some potential in this model, although other models, such as DSS, or IL10-KO or other genetic models should be used in future work to make a complete case for the effectiveness of this agent.
This drug appears to work well. Tell me how you plan to advance the work to get it into a clinical trial. One issue is administration. How is the agent normal taken as a natural compound? I see that the pure compound is potentially distasteful and smells bad. Also, rats can’t vomit, so this potential reaction to oral administration of humans in a purer form can’t be evaluated. Address some of this in the text.
For GSH/GSSG assays it is better to directly isolate and homogenize tissues in acid. We found formic acid to be good. Other initial isolation media tends to overestimate GSSG. For future reference.
In the description of the administration of agents, was tween 80 the vehicle for fenchone? Make this explicit in the text. Also, add oral “gavage” in section 2.3, line 118-119.
There are numerous typos, one is the title, as noted. From appears as “form’ in several cases in figure legends.
Line 424, Fenchone “nearly” restored, is more accurate.
Delete lines 613 to 615.
Comments on the Quality of English LanguageNeeds some editing.
Author Response
- Fenchone Improves TNBS-Induced Colitis in Rats Through Antioxidant, Immunomodulatory, and Cytoprotective Mechanisms, Examines a Terpene's Ability to Alleviate TNBS Damage to the Rat Colon in Both an Acute and Chronic Setting. The study analyzes a variety of parameters to make the case for the agent's effectiveness. Using a steroid as a comparison is very good. Overall, the work demonstrates some potential in this model, although other models such as DSS, or IL10-KO, or other genetic models must be utilized in future work to fully defend the efficacy of this agent.
The TNBS-induced colitis model in rats is validated in the literature and has a well-elucidated inflammatory profile and involvement of the immune system. However, other models can be evaluated in the future to better elucidate the mechanisms of action of (-)-Fenchone.
This medicine seems to work well. Tell me how you plan to move forward with the work to get it into a clinical trial. One issue is administration. How is a normal agent considered a natural compound? I see that the pure compound is potentially unpleasant and smells bad. Furthermore, rats cannot vomit, so this potential reaction to human oral administration in a purer form cannot be evaluated. Address some of this in the text.
Given that (-)-Fenchone has demonstrated significant efficacy in rat models of TNBS-induced colitis, the next steps will involve a series of preclinical and clinical steps to evaluate safety and efficacy in humans.
Although (-)-Fenchone already has acute and subacute toxicity studies for repeated doses as mentioned above and showed low toxicity and safe use through oral administration over 14 days. New studies are required to expand knowledge regarding the toxicity of this substance concerning chronic toxicity studies in animal models to determine the maximum tolerated dose and identify potential adverse effects. As well as, evaluate the pharmacokinetics and bioavailability of (-)-Fenchone, and then think about possible pharmaceutical forms to test this substance in the clinical phase in humans.
We recognize that (-)-Fenchone, as a natural compound, may have unpleasant organoleptic characteristics, such as a strong odor. This can be challenging for oral administration in humans. Rats, because they cannot vomit, do not provide a complete assessment of the taste and odor acceptability of the compound.
To mitigate these issues, we can plan to develop formulations that can mask taste and odor, such as enteric capsules or coated tablets, and explore microencapsulation of (-)-Fenchone to improve palatability and provide a controlled release of the compound.
Additionally, we can investigate alternative routes of administration, such as rectal suppositories or topical formulations, that can bypass the upper gastrointestinal tract, where taste and odor are more problematic.
By addressing these issues comprehensively, we believe we can move forward with (-)-Fenchone into clinical trials with a clear focus on safety and acceptability for patients.
- For GSH/GSSG assays it is best to directly isolate and homogenize tissues in acid. We found that formic acid is good. Other means of initial isolation tend to overestimate the GSSG. For future reference.
The methodology will be reviewed for future studies.
- In the description of the administration of the agents, was Tween 80 the vehicle for the fenchone? Make this explicit in the text. Also, add oral “gavage” in section 2.3, lines 118-119.
Tween 80 is the (-)-Fenchone vehicle. The text was revised according to the guidelines.
- There are several typos, one of which is the title, as noted. From appears as “form” in several cases in the figure captions.
The errors have been corrected.
- Line 424, Fenchone “almost” restored, is more accurate.
Changed as requested.
- Delete lines 613 to 615.
The lines were deleted.
Reviewer 3 Report
Comments and Suggestions for Authors
A well-designed and performed experiment. Well done to the authors. Some minor revisions are needed to improve the manuscript and bring it up to the publishing level.
1) Line 20: Review the language of this line, or add “Aim” at the beginning of the sentence.
2) Line 24: “(-)-Fenchone” not “(-)-fenchone”. Be careful with the punctuation. The name of the compound needs to be written in the same way at the beginning of the sentences throughout the manuscript. If you decide to use“(-)-fenchone” for the middle of the sentence, then it also needs to be consistent throughout the manuscript.
3) Add further information about (-)-fenchone, like their content in different sources, and any safety/toxicity concerns, if applicable. Add the chemical structure of the compounds as a figure.
4) Line 147: Be careful with adding the year in brackets [] which are the same as the ones used for references.
5) Methods: Mention the blindness of assessors and randomisation.
6) Figures: Don’t use commas for p-values.
7) Tables: Improve the format and layout of the tables.
Comments on the Quality of English LanguageGood overall, minor edits needed.
Author Response
1) Line 20: Revise the language of this line or add “Aim” to the beginning of the sentence.
"Aim" was added to the beginning of the sentence.
2) Line 24: “(-)-Fenchone” and not “(-)-fenchone”. Be careful with punctuation. The name of the compound needs to be written the same way at the beginning of sentences throughout the manuscript. If you decide to use “(-)-fenchone” in the middle of the sentence, then it also needs to be consistent throughout the manuscript.
The term (-)-Fenchone was standardized throughout the manuscript.
3) Add more information about (-)-fenchone, such as its contents in different sources and any safety/toxicity concerns, if applicable. Add the chemical structure of the compounds as a picture.
Toxicity and safety information has been added.
4) Line 147: Be careful when adding the year in square brackets [] which are the same ones used for references.
Citations in brackets have been revised and standardized.
5) Methods: Mention the blindness of the evaluators and randomization.
The animals were randomly distributed among the experimental groups. The visual observers who carried out the macroscopic and microscopic analysis were blinded from the identification of the experimental groups.
6) Numbers: Do not use commas for p-values.
The p-value quotes. have been reviewed.
7) Tables: Improve the format and layout of tables.
The tables have been formatted.
Round 2
Reviewer 2 Report
Comments and Suggestions for Authors
The authors have addressed by remarks adequately.